# High-Throughput Phenotypic Characterization and Diversity Analysis of Soybean Roots (*Glycine max* L.)

**DOI:** 10.3390/plants11152017

**Published:** 2022-08-02

**Authors:** Seong-Hoon Kim, Parthiban Subramanian, Bum-Soo Hahn, Bo-Keun Ha

**Affiliations:** 1National Agrobiodiversity Center, National Institute of Agricultural Sciences, RDA, Jeonju 5487, Korea; parthi@korea.kr; 2Department of Applied Plant Science, Chonnam National University, Gwangju 61186, Korea; 3PG & Research Department of Biotechnology and Microbiology, National College (Autonomous), Tiruchirapalli 620001, Tamilnadu, India

**Keywords:** soybean, root, high-throughput phenotyping, germplasm, diversity, WinRHIZO

## Abstract

Soybean (*Glycine max* L.) is a crop native to Northeast Asia, including China, Korea, and Japan, but currently cultivated all over the world. The National Agrobiodiversity Center in Korea at the Rural Development Administration (RDA) conserves approximately 26,000 accessions and conducts characterizations of its accessions, to accumulate new information. Roots are essential organs of a plant, providing mechanical support, as well as aiding water and nutrient acquisition. Currently, not much information is available in international gene banks regarding root characterization. We studied the root phenotype of 374 soybean accessions, using a high-throughput method. Eight root morphological traits (RMT) were studied and we observed that the surface area (SA), number of forks (NF), and number of tips (NT) had a positive correlation with total length (LENGTH), and that link average length (LAL) and other traits all had a negative correlation. Additionally, the correlation between seed traits (height, width, and 100-seed weight) and root traits was confirmed for the first time in this experiment. The germplasms were divided into three clusters by k-means clustering, and orthogonal projections to latent structures discriminant analysis (OPLS-DA) was used to compare clusters. The most distinctive characteristics between clusters were total lateral average length (LAD) and total lateral average length (DIAM). Cluster 3 had the highest LENGTH, SA, NF, and NF, whereas cluster 1 had the smallest LENGTH, SA, and NF. We selected the top 10 accessions for each RMT, and IT208321, IT216313, and IT216137 were nominated as the best germplasms. These accessions can be recommended to breeders as materials for breeding programs. This is a preliminary report on the characterization of the root phenotype at an international gene bank and will open up the possibility of improving the available information on accessions in gene banks worldwide.

## 1. Introduction

Soybean (*Glycine max* L.) is a well-known crop plant with origins in the eastern parts of Asia, including China, Korea, and Japan, but currently grown and consumed all over the world. It remains a major component of the traditional cuisine in the East and South East Asian regions of the world. The high nutritional value of soybean, at a relatively lower cost, has led to its sustained cultivation for thousands of years. Besides being consumed as food, soybean has a myriad of applications in a number of fields, including, medicine, dentistry, fodder and feed, industrial lubricants and solvents, printing, and even in making crayons [1]. Despite its long standing association with the Asian continent, currently, the majority of soybean production occurs in the west, with the top producers being the US, Brazil, and Argentina, which together contribute up to 82% of the total production [1]. 

As per the report from The United States Department of Agriculture (USDA), the world soybean production in 2020–2021 was 366.23 tons, with an average yield of 2.86 tons per hectare [2]. The productivity of soybean is, however, affected by a number of factors, including drought and high daytime temperatures. In addition, the soil moisture conditions in the field across several soybean regions are developing to be suboptimal for normal crop growth, as well as pod filling [2]. Successful establishment of roots can help to overcome plant abiotic stress and is therefore vital to soybean growth, as well as its yield. 

With the increasing demand on food production, all aspects of plant/agricultural sciences must contribute and work collectively to orchestrate production at a maximum efficiency. Among the plant/agricultural sciences, phenotyping is the non-destructive process of recording the visible distinct external traits of organisms, to be used for a number of purposes. In plant science, this is potentially valuable in breeding, for improving the selection efficiency to short amount of time, as well as in understanding the responses of plants to environmental stimuli and in disease management [3]. Currently, vast amounts of data are available with regards to the genetic information of plants/crops, in contrast to the phenotypic data for crops [4]. This necessitates a need to improve the abundance of phenotypic information, which would be immensely valuable in the future, during processes such as breeding and selection. 

Root phenotyping involves morphological characterization of a root system and its architecture. Root system architecture (RSA) is a powerful indicator, which reflects the nutrient status of a plant, as well as its responses to external stimuli [5]. Selection based on root phenotype during breeding can vastly improve the likelihood of producing nutrient efficient crops [6]. Moreover, root phenome has also been correlated to the rhizosphere and root associated microbiome, which indirectly influence plant health and productivity [7]. All these factors stress the importance of expanding the available knowledge on root phenome, since technology is developing rapidly compared to the rate of data accumulation. For example, the relatively newer field of functional phenomics aims to combine physiological data, phenotyping data, and computational biology, to understand various aspects of plant functioning, whereas the quantity of phenotypic data available on crops is comparatively less [8]. 

Studying the root phenome includes understanding the root system architecture (RSA), using several parameters collectively called the root morphological traits (RMT). These parameters include, but are not limited to, root length, average diameter, root surface area, the link average length, and link average diameter [9]. Understanding the root phenome of plants, such as wheat and soybean has started gaining attention and recent works have illustrated the huge potential of root phenome information [9,10]. RSA data can serve as a valuable resource to understand the variations in plant growth and yield in response to different conditions, such as climate change, soil properties, etc., but progress is currently limited by the phenotyping bottleneck. This can be solved by accumulating large data sets of RSA traits or RMTs.

In this study, we measured the root morphological traits (RMT) of soybean landraces originating from China, South Korea, North Korea, and Japan, and analyzed the diversity among germplasms. In addition, we wanted to select useful germplasms and provide the information to breeding programs.

## 2. Results

### 2.1. Variability of Root Morphological Traits

In this study, 374 soybean accessions were analyzed for eight RMTs. All the studied traits showed significant variations among the soybean accessions. The total length (LENGTH) ranged from 703.33 to 3300.00 cm, with a mean of 1759.56 cm. The top 10 accessions based on LENGTH indicated an average of 2649.26 cm. Whereas, the bottom 10 accessions had an average root length of 976.88 cm. The highest and lowest LENGTH was observed in IT208321 and IT162486, which originated from Japan and China, respectively. The majority of the accessions were found to fall between 1200 to 2000 cm (Figure 1). The surface area (SA) values ranged from 124.15 to 524.13 cm^2^, with a mean of 292.84 cm^2^. The top 10 accessions had a combined average of 438.41 cm^2^, against the lowest 10 SA accessions which had an average of 159.77 cm^2^. The highest values for SA were observed in IT208321 (Japan) and the lowest in the accession IT162486 (China). 

The range of average diameter (DIAM) was found to have a relatively even distribution across the accessions and ranged from 0.410 to 0.680 mm, with a mean diameter of 0.536 mm. The top 10 accessions had a DIAM over 0.63 mm. Whereas, the bottom 10 accessions were less than 0.45 mm. The highest DIAM was observed in IT242607 (Korea) and the lowest in IT284095 (Japan). Studying the number of tips (NT) indicated the values were spread across a range of 623 to 3400 NTs, with a mean of 1781.72 mm. The top 10 accessions had average NT of 2974.00, and the bottom 10 accessions showed an average NT of 921.33. The highest NT was observed in IT208245 (Korea) and the lowest in IT162486 (China). The range of values for the number of forks (NF) were from 1180 to 11638.33 mm, with a mean of 5226.12. Most of the accessions had 3200 to 5200 forks and gradually decreased on both sides. The top 10 accessions had an average NF of 8948.67 compared to the bottom 10 accessions, which had NG values almost four times less at 2239.67. The highest area of NF was observed in IT 216313 (Korea) and the lowest in IT 162486 (China). For LAL, the values ranged between 0.150 and 0.380 cm, with a mean of 0.217 cm and a large number of plants showed LAL values at around 0.2 cm (Figure 1). However, a more even distribution was observed in the case of the LAD values, which ranged between 0.39 and 0.73 mm. 

Based on the skewness and kurtosis, seven traits (LENGTH, SA, DIAM, NT, NF, LAL, LAD, and LABA), except for LABA, were normally distributed (Table 1). However, LABA showed the highest CV (59.67%), followed by NF (34.33%), NT (30.59%), LEGNTH (24.81%), and SA (23.84%) (Table 1). The RMT parameters, including NT, NF, LENGTH, SA, and LAL, showed a high CV (>10%), demonstrating significant differences among the accessions. The factors DIAM and LAD showed the lowest CV, of around 10%, and showed a normal distribution among the accessions. The skewness of the DIAM and LAD values were near to zero, indicating an almost normal distribution of these parameters across the accessions. The LAL data from the accession showed the highest positive skewness value, indicating a skewing of the distribution towards the left. Therefore, most of the accessions had a lower LAL in general. A negative skewness value was observed in the case of LABA, indicating a skewing of the distribution towards the right, which denotes an increasing trend of LABA across the accessions (Table 1). An analysis of variance (ANOVA) was performed, to identify differences between germplasms and between replications (*n* = 3) within a germplasm. It can be observed that all the studied RMTs were distinctly different among the accessions studied, which were also statistically significant, but homogenous within the replicates (Table 2). Therefore, it was found that significant diversity could be observed between the 374 germplasms, but not within the germplasms.

### 2.2. Correlation Analysis

A correlation analysis was conducted, to examine the association between the studied RMTs (Figure 2). LENGTH showed significant correlations with SA, DIAM, NT, NF, LAL, and LABA (*p* < 0.001). Among these, length and SA showed a highly significant positive correlation (r = 0.94, *p* < 0.001) with each other. This was expected, as longer roots tend to have a greater area/volume. The traits NF and NT also showed a high correlation with LENGTH (r = 0.92, r = 0.88, *p* < 0.001, respectively). This indicates a well-established fibrous nature of the root, with more soil coverage, thereby improving the nutrient acquisition. Traits of SA and NF were also found to exhibit a high positive correlation with NT (r = 0.92, r = 0.84 *p* < 0.001, respectively). This can also be understood to be an efficient growth and development of the soybean roots. The positive correlation between DIAM and LAD (r = 0.93) indicated that the lateral roots were also significantly well developed, similarly to the main root. LAL showed a negative correlation with NF, SA and NT, and length (r = −0.76, r = −0.60, r = −0.59, r = −0.56, *p* < 0.001). The parameter LABA was found to have a low correlation with all the other studied RMTS.

### 2.3. Phenotypic Clustering and Diversity Analysis

Among the eight principal components, three principal components with an eigenvalue of greater than 0.942 could explain a cumulative proportion representing the total variance of 90.45%. The first principal component had 4.24 traits and could explain 53.00% of the total variation. The second principal component included 2.05 traits, which accounted for 25.67%, and the third principal component included 0.94 traits, which could explain 11.78% (Table 3). The first principal component (PC1) had a significant positive correlation, in the order of NF (0.472), length (0.467), and SA (0.448); whereas, it had a significant negative correlation with LAL (−0.361). The second principal component (PC2) had a significant positive correlation in the order DIAM (0.661) and LAD (0.659), whereas a negative correlation was observed with LAL (−0.170) and NT (−0.110). A highly significant positive correlation was observed with LABA (0.987) in the third principal component (PC3). With PC1, traits related to the root length were highly correlated, whereas with PC2, the traits related to the root diameter were highly correlated to each other. In addition, the trait related to root system architecture (RSA) was related to PC3 (Table 4). 

The RMTs and RSA were found to segregate into three clusters, when k-means clustering was carried out, and this is shown in the dendrogram (Figure 3a). A total of 73 and 97 accessions were found to be grouped in cluster 1 and cluster 2, respectively, and 204 accessions, including control (Enrei), were found in cluster 3 (Figure 3a). A scatter plot of the RMTs adjusted according to PC1, PC2, and PC3 is shown in Figure 3b. On the right side of the vertical axis of the plot, accessions which showed high values for LENGTH, NF, SA, NT, and LABA were found to be grouped together, and many of the germplasms previously included in cluster 1 were related to these traits. While on the left side, accessions that initially showed high values for the traits LAL, DIAM, and LAD were found to be grouped together. Accessions with large DIAM and LAD were distributed above the horizontal axis, indicating that the germplasms with thick primary and lateral roots may have had well-established roots, as LAL was also found to group with DIAM and LAD. Germplasms with large LENGTH, NT, and LAD grouped below the horizontal axis, which indicated the distribution of germplasms with long whole roots, lateral roots, and many root endings. The clusters were then compared using orthogonal partial least squares discriminant analysis (OPLS-DA). The major variables for distinguishing clusters were (in order of importance) LAD, DIAM, NF, SA, LENGTH, and NT (Figure 3c). The most distinctive characteristics between the adjacent clusters were LAD and DIAM. The larger the LAD and DIAM, the parameters evaluating the root thickness among the RMTs, the higher the ability of the root to penetrate the soil layer, which is advantageous for extending roots in search of water. Based on eight RMTs, we divided them into three clusters and compared them with a boxplot. A high variability was confirmed for the eight RMTs across the accessions. Cluster 3 was the largest cluster, with n = 204, and constituted accessions with high values for five RMTs (LENGTH, SA, NT, NF, and LABA). The high distribution could be attributed to the large number of accessions (n = 204), but the median was found to be distinctly higher than the other two clusters. Among the RMTs, the total length of a root (LENGTH) is known to play an important role, such as adaptation to environmental changes [11]. On the other hand, cluster 2 (n = 97) indicated a high root thickness (DIAM, LAD). In cluster 1, the LAL variable was particularly high (Figure 4 and Figure 5). The lateral roots (LAL) can be directly related to the absorption of water and nutrients by the plants [12]. 

### 2.4. Correlation Between Root and Seed Traits

Soybean seeds are directly related to yield, but few prior studies have been conducted on the correlation between root and seed traits. In this experiment, the association between eight root phenotypes and seed traits (height, width, and 100-seed weight) was confirmed. First of all, the correlation between height of seed and root SA and LAD was confirmed (r = 0.48, r = 0.44 *p* < 0.001, respectively). In addition, the correlation between 100-seed weight and SA, width of seed, and length was confirmed (r = 0.52, r = 0.30, *p* < 0.001, respectively) (Appendix A).

## 3. Discussion

Roots represent a vital part of plants, owing to their major role in water and nutrient absorption, as well as in providing physical support to the plants. An optimised development of the roots is, therefore, a prerequisite for the sustained productivity of the plant under varying conditions and challenges in the field. However, they remain comparatively difficult to study compared to other parts of the plant, due to the diffculty in continuously accessing the root during the process of plant growth and development. With the recent development of technology, such as magnetic resonance and three-dimensional (3D) computed tomography, progress in agricultural research of root systems has seen a tremendous increase. However, such state-of-the-art equipment is often too expensive, and excavating the roots without damaging them remains a challenge. 

In order to measure the RMTs of a large amount of samples, such as in the germplasms conserved by a genebank, a method of phenotyping a large amount of resources, while minimizing root damage, is required. For this reason, we used polyvinyl chloride (PVC) columns, which have been used by several researchers previously [13]. The quantity of data collected using traditional methods was highly limited compared to the newer methods that can carry out two-dimensional analysis of high-resolution images using software packages [14,15]. Further developments have also been made, but the 2D analysis of RMTs remains a comparatively cheap and efficient mode of plant phenotyping. Several sofware packages are available for RMT analysis, which can be chosen between based upon the study and parameters one wishes to analyse [15]. In the present study we used “WinRhizo”, which provides the fastest and most accurate estimation of root phenotype and has been extensively used to study legume roots [13]. 

Across the studied accessions, a high coefficient of variation (CV) of more than 30% was observed for LABA, NF, and NT, indicating they were the most diverse RMTs and can be used to observe difference aross the accessions. A root phenome study among wild adzuki beans showed that highest CV among the RMTs were found to be NT (59.37%), SA (56.42%), and LENGTH (51.91%) [14]. In another study on mungbeans, root volume (RV) and root tips or number of tips (RT/NT) showed high CV among the studied population [16]. Even a single parameter, such as total root length measured at seedling stage of winter oilseed rape, can help to predict the nitrogen uptake and seed yield [17]. RMTs, can therefore serve a number of purposes, including seggregation of plants and prediction of yield, which can be very useful for breeding. Statistical analysis of our data using ANOVA indicated that our results showed a clear seggregation of germplasms compared to previous research conducted using the same methods [13,14]. In our study, all eight RMTs were found to be significantly different between the germplasms, and there was homogenecity among the three replications within each accession, whereas the previous study indicated two RMTs to be significantly variable across the studied germplasms (Table 2) [13,14]. In the present experiment, soybeans were grown in fields for two consecutive years (2019–2020), heterogeneous germplasms within accessions were removed according to agronomic characteristics (UPOV guidelines), and resources with a high seed vitality, as well as uniformity, were used. During PCA analysis, since the cumulative contribution rate of the PC1 and PC2 accounted for 78.68% of the total, the two main components were considered to be very important main components for comparing the phenotypic diversity of soybean landrace roots. In general, LENGTH, SA, NF, and NF were highly positively correlated, and LAL was negatively correlated, contributing to one principal component. In addition, the second principal component had a high positive correlation with DIAM and LAD. In particular, these six root traits are considered to be important traits for classifying soybean landraces.

A recont study reported that LENGTH, SA, and LAD had higher variability than DIAM, LAL, and LABA. In particular, it was emphasized that the Length and SA of North Korean germplasms have a higher variability than germplasms originating from other countries [13]. In the current results, for germplasms originating in North Korea, only LAL confirmed a high variability, whereas germplasms originating in Korea showed the highest variability for the six RMTs (LENGTH, SA, DIAM, NT, NF, and LAD) (Appendix A). This may be due to the high seed vitality, through continuous cultivation for 2 years (2020–2021). Correlation analyses among the RMTs often resulted in mixed or ambigious relationships with each other. In the current study, strong positive correlations were observed between LENGTH-SA, LENGTH-NT, LENGTH-NF, SA-NT, SA-NF, DIAM-LAD, and NT-NF pairs. Strong negative correlations were observed between NF and LAL. In a earlier study on soybean roots [18], the total root length was highly corelated to the total surface area (r^2^ = 0.93). A recent study on soybean [19], reported that a highly positive corelation was observed for total root length with SA, RV (root volume), and LABA, among the RMTs; as well as, surface area with root volume, among the RMTs studied. Total root length has also been reported to have a positive correlation with projected area (surface area) and forks; projected area with forks and total lateral length; and main total length with total lateral length [9]. In another study of wheat root phenome, the root area was found to be positively correlated with root volume [20]. In general, the RMT root length is found to be positively correlated with surface area and root volume. Trends in such RMT correlations can help to understand the overall root development. Although, some reports exist where some RMTs exhibited contrasting trends [14]. This becomes much more complex when we start comparing across other crop plants [14]. 

Among genetic resources (cultivar, wildtype, landrace etc), landraces have long been recognized for their importance as a source of local adaptation, stress tolerance, yield stability, and seed nutritional properties, and they can be utilized as an excellent material in breeding programs [21]. We ranked the accessions based on RMTs, to select the best soybean landraces in this experiment. IT208321 (Kanro) had the longest LENGTH, and the SA and NF were ranked in the top 10 accessions. This accession was originally collected in Japan by the Vavilov Institute of Plant Genetic Resources (VIR), which was introduced and conserved by NAC from 1997. Its seeds are large, and it is a resource used in root research and disease resistance research [22]. Following this, IT216313 (Sokcheong) had the second longest LENGTH, and its SA and NT were included in the top 10 accessions. This accession was collected by NAC in 2001, in Ganghwa-eup, Gangwon- Province, Korea. It has a fast maturation period and a high production. In particular, the 100 seed weight was 38.8 g, and it is one of the most popular breeding materials, owing to its large seed size [23]. The third longest LENGTH was IT216137, with its root thickness-related traits (DIAM and LAD) ranked in the top 10. This accession was collected by NAC, in 1999 on Jeju island, Korea [24]. Studying the RMTs of the accessions by grouping of origin confimed similar pattern of root system development (Appendix A).

Cluster analysis revealed that the top 10 accessions in four traits (LENGTH, SA, NT, NF) were included in cluster. A high LENGTH has the advantageous characteristic of being able to absorb deep soil water from the deep soil layer, where the roots can grow deep in the soil and use the water [25]. In the past, accessions with high SA were found to have increased tiller number, and this is directly attributable to the increase in biomass of the shoot [26]. Accessions with the top 10 root thicknesses (DIAM, LAD) were included in Cluster 2. On the other hand, the top 10 LAL were distributed among cluster 1 (n = 7) and cluster 2 (n = 3), and the top 10 accessions have high values for LABA were spread across cluster 1 (n = 2), cluster 2 (n = 1), and cluster 3 (n = 7), respectively (Figure 4). It will be interesting to find the genes responsible for these RMTs in the recommended germplasms, which could be used in the future to develop drought resistant or flood resistant resources.

Drought is a threat to soybean production worldwide, and soybean roots and nodules are important indicators of drought [27]. Enrei is a Japanese germplasm well documented for its high-nodulating efficiency [28,29]. The measurement of root nodules using Winrhizo software had a poor accuracy, so prior results were analyzed by applying machine learning [30]. In the current study, we directly measured the number of root nodules in this experiment, and the average number of Enrei was found to be 40. The germplasm that exhibited more root nodules than Enrei was IT162251 (53 nodules), followed by IT162023 with 47 nodules. The RMT was found to be greater for SA, DIAM, and LAD compared to Enrei. These two accessions were collected by the United States Department of Agriculture (USDA) in 1930 from Jagang Province in North Korea and Hokkaido Island in Japan, respectively [31]. These two germplasms are recommended as excellent materials for drought-related research and breeding programs. 

## 4. Materials and Methods

### 4.1. Soybean Germplasms and Growth Conditons

In this study, 374 soybean landraces were obtained from the National Agrobiodiversity Center (NAC) of the Rural Development Administration (RDA), Korea. The material of 374 resources, including a control (Enrei), was cultivated at the NAC. The collected accessions were spread across four countries, including South Korea (115), North Korea (60), China (98), and Japan (100). To maintain purity, the soybeans germplasms were grown in fields for two consecutive years (2019–2020); heterogeneous germplasms within accessions were removed, according to agronomic characteristics (UPOV guidelines); and resources with high seed vitality and uniformity were used. Enrei is one of the representative varieties of Japan and has the characteristic of having a lot of root nodules [19]. Each accession was grown in horticultural soil (Tobirang, Baekkwang Fertility, Korea) up to 38 cm, in three Polyvinyl chloride (PVC) columns measuring 8 cm in diameter and 40 cm in height, with three replications (Table 5). Commercial horticultural soil was used for easy extraction and to reduce damage to the roots during excavation, as it facilitates easy separation of the roots and the soil. The physicochemical properties of the soil used are given in Appendix A. Each bottom was covered with a mesh, to allow movement of water, and the meshes were fixed with cable ties. The research was conducted under controlled environmental conditions (21 ± 1 °C) in greenhouse installed at the NAC, and plants were grown for 40 days (23 April to 2 June 2021) with three replicates for each accession (Table 5).

### 4.2. Phenotypic Data Collection and Rroot Morphological Traits (RMTs)

Root excavation was performed during second trifoliate leaves stage (V2), and the soil was completely removed. Documentation of the roots and analysis of RMTs were carried out as reported previously [25]. In short, roots were gently shaken to remove adhering soil particles, washed carefully by hand after placing them on top of a sieve and immediately moved for scanning. Scanning of the roots was performed on an Expression 12,000XL (Epson; Nagano, Japan), by placing the roots inside a transparent plastic tray (30 cm long × 20 cm wide) containing clean tap water [25]. A total of eight traits (LENGTH, SA, NF, DIAM, NT, NF, LAL, LAD, and LABA) were measured using WinRHIZO Pro software (Reagent Instruments, Quebec, Canada). A description of the extracted root morphological traits is shown in Figure 6.

### 4.3. Seed Phenotypic Data Collection

We used the seed phenotype measurement program developed in previous experiments to measure the height, width, and thickness of seeds, according to the UPOV guideline [32,33]. One hundred grains for each germplasm were measured, and the average value was used. Then, 100-seed weight was measured with an electronic scale (OHAUS^TM^—PR224, Parsippany, NJ, USA) at the NAC, under limited temperature (10 °C) and humidity (30%) conditions, in 3 repetitions.

### 4.4. Statistical Analysis

We used the Bartlett sphericity test to see whether each variable was independent from the others. In addition, a Kaiser−Meyer−Olkin (KMO) test was performed to confirm the rationality of the data structure. Histograms were prepared in Microsoft Excel (2016 package), and an analysis of variance (ANOVA) carried out using SAS 9.4 (SAS, Gary, NC, USA). Principal component analysis was performed using XLSTAT software v2019 (Addinsoft, Paris, France). The Pearson’s correlation coefficient between root agronomic traits was calculated using the "corrplot" package, in R v4.1.0. The k-means cluster analysis, principal component analysis (PCA), and orthogonal partial least squares discriminant analysis (OPLS-DA) were obtained for comparison between clusters using SIMCA software (ver.13.3, Umetrics, Umea, Sweden). Boxplots were generated using plotly chart studio (https://chartstudio.plot.ly, accessed on 19 June 2022).

## 5. Conclusions

In this study, we measured eight root morphological traits (RMTs) of 374 soybean landraces originating from China, Korea, North Korea, and Japan. As far as we know, this is the first time Chinese, South Korean, North Korean, and Japanese landraces have been studied for their root pheonotypes by growing PVC, followed by scanning and software-based analysis. [34]. This data will augument the available information on the conserved soybean accessions in the NAC collection. When comparing with the previously published RMT data on soybean germplams, our results showed a staistically significant homgenecity between the replications and heterogencity between the studied accessions. Correlation analyses confirmed that LENGTH was the RMT exhibiting the highest level of correlations with the other studied RMTs. Vandamme and coworkers also reported root length to be a statistically significant RMT, which varied between germplasms in the early stage of soybeans [35]. We also observed a trend of a negative correlation between LAL and the remaining seven traits. Clustering analyses revealed that the accessions seggregated into three distinct clusters, based on their RMTs. We have reported the top ten accessions for each of the studied RMTs and also the high nodulation in comparison with Enrei soybean plants, which can be used for selective breeding. Future research, such as the mapping of quantitative trait locus (QTL) using a genome-wide association study (GWAS), can help us to understand the genomic perspective of the exhibited RMTs.

## Figures and Tables

**Figure 1 plants-11-02017-f001:**
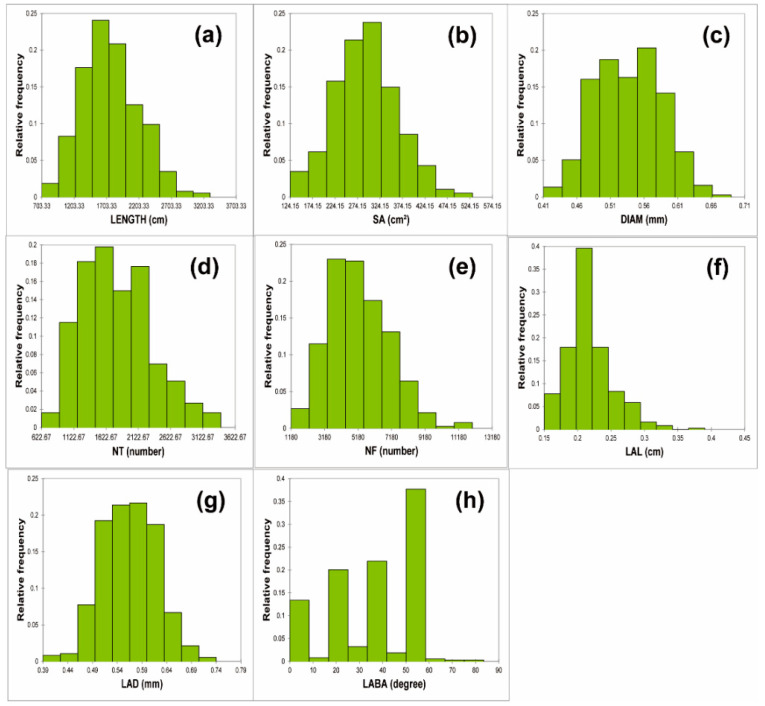
Histogram of root morphological traits (RMT). Each figure shows (**a**) total length (LENGTH); (**b**) surface area (SA); (**c**) average diameter (DIAM); (**d**) number of tip (NT); (**e**) number of forks (NF); (**f**) link average length (LAL); (**g**) link average diameter (LAD); and (**h**) link average branching angle (LABA), respectively.

**Figure 2 plants-11-02017-f002:**
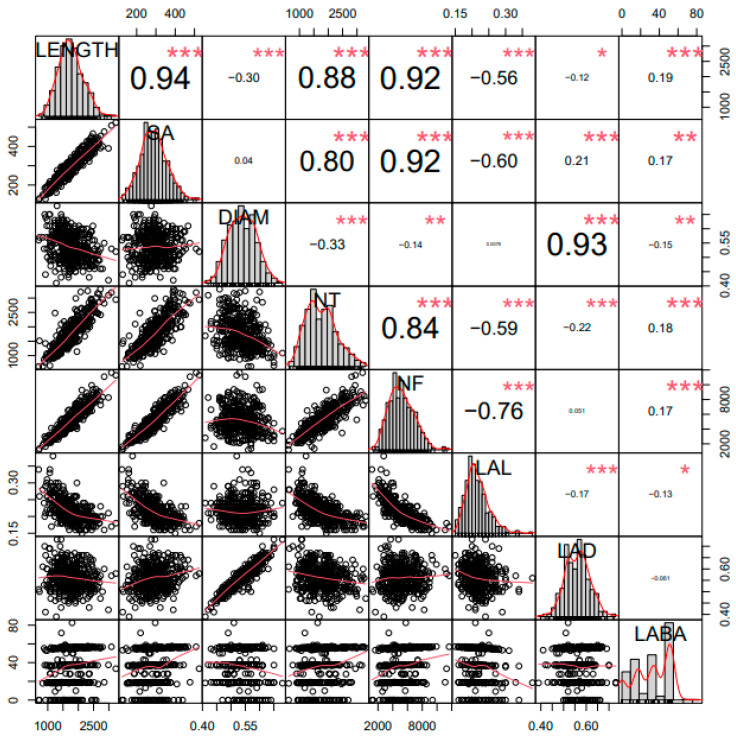
Correlation analysis of eight root morphological traits. *** Significant at the 0.001 level of probability, ** significant at the 0.01 level of probability, and * significant at the 0.05 level of probability.

**Figure 3 plants-11-02017-f003:**
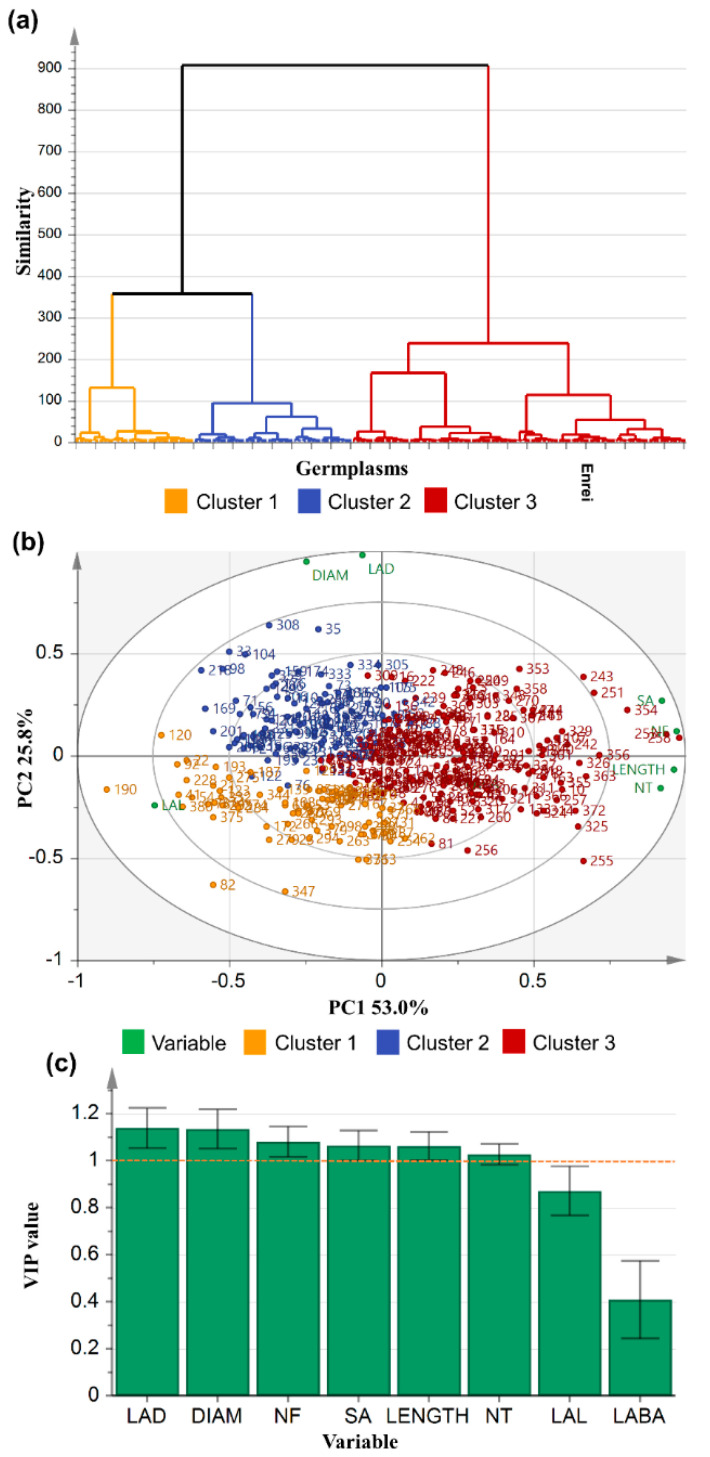
Denerogram classified into three clusters by k-means (**a**) and distribution of eight root morphological traits variable and 374 germplasms of a scatter plot of principal component 1 (53.0%), 2 (25.79%), and 3 (11.78%) (**b**). Orthogonal projections to latent structure discriminant analysis (OPLS-DA) classified into three clusters by k-means ananlysis. (**c**) Variable importance in the projection (VIP) value.

**Figure 4 plants-11-02017-f004:**
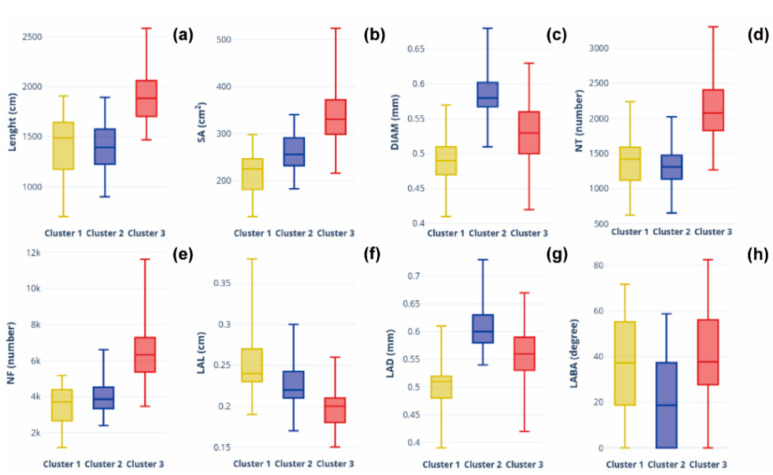
Box plot of eight root morphological traits between three clusters separated by k-means. (**a**) Total root length (LENGTH); (**b**) total surface area (SA); (**c**) total average diameter (DIAM); (**d**) number of tips (NT); (**e**) number of forks (NF); (**f**) link average length (LAL); (**g**) link average diameter (LAD); (**h**) link average branch angle (LABA).

**Figure 5 plants-11-02017-f005:**
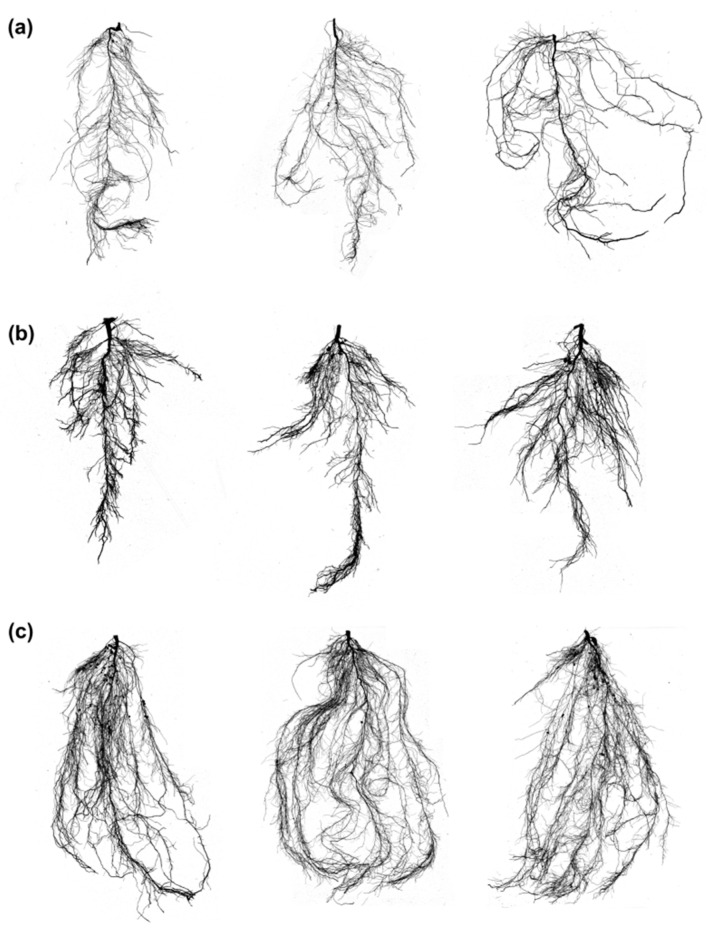
Clusterwise root phenotypes (**a**) Cluster1—highlighting high LAL (IT160933, IT160934, and IT162486); (**b**) Cluster 2—highlighting thicker roots with DIAM and LAD (IT24260, IT160839, and IT160851); (**c**) Cluster 3—roots with high LENGHTH, SA, NT, and NF (IT208321, IT216313, and IT285979).

**Figure 6 plants-11-02017-f006:**
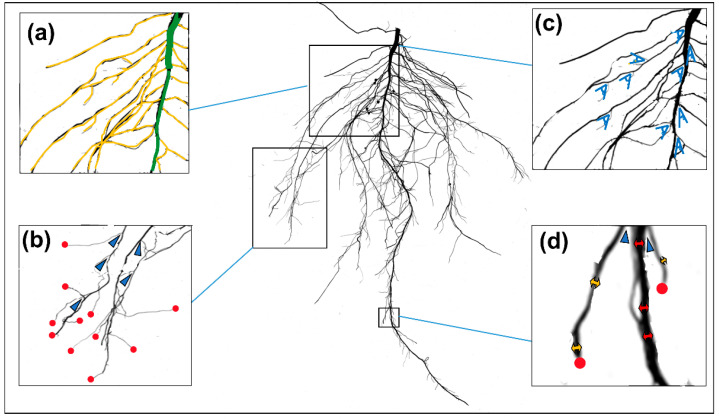
The figure shows the root morphological trait of the control (Enrei). (**a**) Total length (LENGTH) is the sum of the main root (green) and total lateral average length (LAL, yellow), and SA is the sum of the total surface area; (**b**) tip (red) is the root ending and fork (blue) is the new branching incidences. Each number has been abbreviated as NT and NF, respectively; (**c**) link average branch angle (LABA) measures the angle of the lateral roots (blue) from the main root; (**d**) average total diameter (DIAM) was measured for the main root diameter (red double arrow) and the lateral root diameter (yellow double arrow). In addition, the lateral average diameter (LAD) was measured at the lateral root (yellow double arrow).

**Table 1 plants-11-02017-t001:** Descriptive statistics for root morphological traits in 374 soybean germplasms.

Traits	Range	Mean	SD ^a^	CV (%) ^b^	Skewness	Kurtosis
LENGTH	703.330–3228.240	1758.653	436.312	24.809	0.325	−0.027
SA	124.150–524.130	292.745	69.800	23.843	0.244	0.006
DIAM.	0.410–0.680	0.536	0.048	9.038	0.079	−0.446
NT	622.670–3305.670	1779.992	544.509	30.591	0.451	−0.271
NF	1180.000–11638.330	5224.855	1793.769	34.331	0.489	0.154
LAL	0.150–0.380	0.217	0.034	15.522	1.002	1.881
LAD	0.390–0.730	0.562	0.056	9.898	0.064	−0.014
LABA	0–82.510	35.476	20.107	56.678	−0.416	−1.060

**^a^** Standard deviation; **^b^** Variation coefficient.

**Table 2 plants-11-02017-t002:** Analysis of variance (ANOVA) with eight root morphological traits.

Traits	Source	DF	Type III SS	Mean Square	F Value	Pr > F
TRL	Accession	373	210688645.7	564848.9	3.73	<0.0001
	rep	2	157269.4	78634.7	0.52	0.595
SA	Accession	373	5399809.086	14476.7	4.14	<0.0001
	rep	2	3760.97	1880.485	0.54	0.5839
AD	Accession	373	2.57849663	0.00691286	4.56	<0.0001
	rep	2	0.00334633	0.00167316	1.1	0.332
NT	Accession	373	328293648.4	880143.8	4.64	<0.0001
	rep	2	271965.2	135982.6	0.72	0.4887
NF	Accession	373	3551476,034	9521383	5.12	<0.0001
	rep	2	3296362	1648181	0.89	0.413
LAL	Accession	373	1.25066563	0.00335299	3.22	<0.0001
	rep	2	0.00380687	0.00190343	1.83	0.1616
LAD	Accession	373	3.40960212	0.00914102	4.6	<0.0001
	rep	2	0.00314071	0.00157036	0.79	0.4541
LABA	Accession	373	446569.9132	1197.2384	2.36	<0.0001
	rep	2	1189.8771	594.9385	1.17	0.3096

**Table 3 plants-11-02017-t003:** Eigenvalue and proportion of principal components for the eight RMTs.

Principal Component	Eigenvalue	Variability (%)	Cumulative (%)
**1**	**4.241**	**53.007**	**53.007**
**2**	**2.053**	**25.669**	**78.676**
**3**	**0.942**	**11.776**	**90.452**
4	0.512	6.402	96.854
5	0.167	2.082	98.937
6	0.041	0.508	99.445
7	0.040	0.494	99.939
8	0.005	0.061	100.000

**Table 4 plants-11-02017-t004:** Three principal components among eight the RMTs.

Traits	Eigenvectors of Principal Component		
	1	2	3
LENGTH	**0.467**	−0.050	−0.063
SA	**0.448**	0.185	−0.029
DIAM	−0.118	**0.661**	0.046
NT	**0.446**	−0.110	−0.077
NF	**0.472**	0.081	−0.058
LAL	−0.361	−0.170	0.024
LAD	−0.029	**0.685**	0.096
LABA	0.123	−0.096	**0.987**
Eigenvalue	4.241	2.053	0.942
Variability (%)	53.007	25.669	11.776
Cumulative (%)	53.007	78.676	90.452

**Table 5 plants-11-02017-t005:** Experimental details and design of the present study.

Cultivated Period	Environment	Soil	Pot	Design	Control
40 days (23 April to 2 June 2021)	Controlled Greenhouse (21–22 °C)	horticultural soil (Tobirang, Baekkwang Fertility, Korea)	PVC pipe (40 × 8)	Randomized block design (3 reps.)	Enrei

## Data Availability

Not applicable.

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
