# Peer review of "High-Throughput Phenotypic Characterization and Diversity Analysis of Soybean Roots (Glycine max L.)"

_plants, 2022, doi:10.3390/plants11152017_

Round 1

Reviewer 1 Report

In the reviewed work entitled 'High-Throughput Phenotypic Characterization and Diversity Analysis of Soybean Roots (Glycine max L.)' root phenotype of 374 soybean accessions was analyzed using a high-throughput method. Eight root morphological traits (RMT) were studied. The authors selected the top 10 accessions that they recommend for use in breeding work for this species.

There are many stylistic and grammar errors in the manuscript. Some literature is incorrectly cited. I corrected these errors in the attached file.

The introduction is properly written. The methodology could include more details. The methods used were not well described and not explain how the experiment was conducted. From the plant, we suddenly move on to data analysis. How was the data obtained?

The description of the results is correct.

I don't like the discussion, especially the last paragraph which should be redrafted. The same goes for the conclusions. The authors have not checked these parts of the text, and in both parts, there are sentences that do not make sense.

The work is mediocre, but it would surely increase its value if the roots were assessed at several time points. Taking into account how little effort was put into obtaining results, obtaining additional data from other plant growth stages or under stress conditions would definitely increase the value of the work.

Author Response

The reviewer's advice and meticulous attention were able to advance my paper one step further. Q1) The introduction is properly written. The methodology could include more details. The methods used were not well described and not explain how the experiment was conducted. From the plant, we suddenly move on to data analysis. How was the data obtained?   A1) Thank you for your insight. The methodology is now amended with description of the experimental plan, sampling and initial processing of the root samples. Q2) I don't like the discussion, especially the last paragraph which should be redrafted. The same goes for the conclusions. The authors have not checked these parts of the text, and in both parts, there are sentences that do not make sense. A2) We thank the reviewer for this comment. Discussion and conclusions have now been thoroughly redrafted according to the reviewer’s comments.

Q3) The work is mediocre, but it would surely increase its value if the roots were assessed at several time points. Taking into account how little effort was put into obtaining results, obtaining additional data from other plant growth stages or under stress conditions would definitely increase the value of the work.

A3) We understand and accept the reviewer’s comments. We will definitely incorporate them in our present and future studies on root phenotyping. Thank you very much.

Reviewer 2 Report

The revised paper "High-Throughput Phenotypic Characterization and Diversity Analysis of Soybean Roots (Glycine max L." by Seong-Hoon Kim and co-authors focuses on interesting biological problems regarding plants' phenotypic diversity, particularly root phenotypic diversity. 

The root diversity includes two main characteristics: morphology and architecture of the root system (RSA). The measurement and data about root system architecture (RSA) provide an indicator reflecting the nutrient absorption ability of the plant and its responses to external factors. The root architecture and development are related to the rhizosphere and root-associated microbiome, which indirectly influence plant performance. Importance expanding the available scientific biological knowledge on root phenome is required. The Authors mentioned the field of functional phenomics, which combines physiological data, phenotyping data, and computational biology to understand various aspects of plant functioning. In contrast, the amount of phenotypic data available is scars.

It is worth mentioning that the studies covered root morphological traits (RMT), including the surface area (SA), number of forks (NF), and number of tips (NT), which had a positive correlation with total length (LENGTH). At the same time, the average size (LAL) and other traits are negatively correlated. 

The revised study presents a large body of performer measurements and analyses. 

However, there is a severe lack of clearly highlighted scientific problems and presentation of the scientific hypothesis verified in this study.

At the current stage, the paper is a descriptive agricultural tool. Agriculture can apply the results of scientific research. It is essential to distinguish between science, the theoretical basis, biological phenomenon (such as the ability of plants to adapt and resulting phenotypic variety), and the application of these natural processes. 

It is also of great concern that the author mention in the paper about breeding and GM (genetically modified) crops. GM food is at least controversial or forbidden in some countries. It should be sincerely discussed.

Author Response

The reviewer's advice and meticulous attention were able to advance my thesis one step further. Q1) However, there is a severe lack of clearly highlighted scientific problems and presentation of the scientific hypothesis verified in this study.

A1) Thank you very much for your comment. We have tried to improve highlighting the scientific problems and hypothesis in the end of the introduction.

Q2) It is also of great concern that the author mention in the paper about breeding and GM (genetically modified) crops. GM food is at least controversial or forbidden in some countries. It should be sincerely discussed.

A2) Again, we thank the reviewer for his insight with regards to GM crops. As a plant germplasm collection of Rural Development Administration (RDA) based in South Korea, it would be against our institute (Genebank) policy to discuss on GM crops and we have therefore removed any reference with regards to mentioning of GM in our manuscript. GM crops are however discussed in detail by our sister institutes of the RDA.

Reviewer 3 Report

The paper suffers from some important weaknesses:

1) Very few root traits. The main aim of the paper is the root phenotyping of soybean. But phenotyping is a high-throughput method applied in order to provide measurements for several root traits, not eight! Few root traits did not permit a depth evaluation of the root system, especially for breeding. For example, the root length ratio, root fineness, root tissue density, etc, pointed out more functional roles than root length. Further, the average diameter is a poor functional parameter. It is better to use the root fineness (Ryser, P. Intra- and interspecific variation in root length, root turnover and the underlying parameters. In Inherent Variation in Plant Growth: Physiological Mechanisms and Ecological Consequences; Lambers, H., Poorter, H., Van Vuuren, M.M.I., Eds.; Backhuys Publishers: Leiden, The Netherlands, 1998; pp. 441–465) and/or the diameter classes.

2) Lacking of the root phenotyping finalization. Usually, the root phenotyping is accompanied by agronomic data (for example, root vigorous correlated with high production, etc.) or stress tolerance/sensitivity (for example, “steep, cheap and deep” architecture correlate with high productivity in drought and nutrient deficiency soils, etc) or genomic data for gene identification (for example, as Guo et al 2021, the reference cited by Authors).

The Authors indicated that the “….Soybeans were grown in fields for two consecutive years (2019-2020), heterogeneous germplasms within accessions were removed according to agronomic characteristics (UPOV guidelines), and resources with high seed vitality and uniformity were used…”. Have you agronomic data? If the response is yes you can: 1) correlate the root morphology with agronomic data, 2) by multivariate analysis, to evaluate the root morphology traits that better correlate with the soybean productivity.

3) Lacking of experimental design. What is the experimental design? Randomization? Blocking? Factorial? This is important because the Authors carried out the experiment in the greenhouse.

3) Details of the root phenotyping throughput. Because the root phenotyping is carried out for 375 soybean genotypes, it is important to provide the root phenotyping throughput…How much root sample for person?

4) Lacking information for the PCA. The Authors used the PCA but the AA did not report the Rotation method, the KMO adequacy and the Bartlelett’s test significance.

5) The conclusion paragraph was not well written. The Conclusion must report a sum up what you have presented in your research without sounding redundant and write the most important points and supporting evidence of your arguments or position of the thesis.

For these reasons, the paper should be reconsider after major Revisions taking in account of my suggestions for the publication in Plants.

Author Response

The reviewer's advice and meticulous attention were able to advance my thesis one step further. Q1) Very few root traits. The main aim of the paper is the root phenotyping of soybean. But phenotyping is a high-throughput method applied in order to provide measurements for several root traits, not eight! Few root traits did not permit a depth evaluation of the root system, especially for breeding. For example, the root length ratio, root fineness, root tissue density, etc, pointed out more functional roles than root length. Further, the average diameter is a poor functional parameter. It is better to use the root fineness (Ryser, P. Intra- and interspecific variation in root length, root turnover and the underlying parameters. In Inherent Variation in Plant Growth: Physiological Mechanisms and Ecological Consequences; Lambers, H., Poorter, H., Van Vuuren, M.M.I., Eds.; Backhuys Publishers: Leiden, The Netherlands, 1998; pp. 441–465) and/or the diameter classes.   A1) Thank you very much for your valuable insight. We understand your point. We have tried to include any other collected data from the study to accompany the root phenotypic data and will incorporate the reviewer’s comments in future studies. Q2) Lacking of the root phenotyping finalization. Usually, the root phenotyping is accompanied by agronomic data (for example, root vigorous correlated with high production, etc.) or stress tolerance/sensitivity (for example, “steep, cheap and deep” architecture correlate with high productivity in drought and nutrient deficiency soils, etc) or genomic data for gene identification (for example, as Guo et al 2021, the reference cited by Authors).

A2) We thank the reviewer for his comment. We have now included description of the experimental plan, sampling and initial processing of the root samples in the methods section.

Q3) Details of the root phenotyping throughput. Because the root phenotyping is carried out for 375 soybean genotypes, it is important to provide the root phenotyping throughput…How much root sample for person?   A3) The comment has now been addressed in the revised methods section   Q4)  Lacking information for the PCA. The Authors used the PCA but the AA did not report the Rotation method, the KMO adequacy and the Bartlelett’s test significance.   A4) The Bartlett sphericity test and the Kaiser-Meyer-Olkin (KMO) test were checked when using XLSTAT for principal component analysis. The results show that the P value of Bartlett's test rejected null hypothesis, that is, it is considered that the research data can be extracted by principal components, and the hypothesis 2 is satisfied. In addition, the KMO test coefficient is greater than 0.6, indicating that the sample meets the requirements of a reasonable data structure. Corresponding text has been added to the manuscript.   Q5) The conclusion paragraph was not well written. The Conclusion must report a sum up what you have presented in your research without sounding redundant and write the most important points and supporting evidence of your arguments or position of the thesis.   A5) We thank the reviewer for this comment. Discussion and conclusions have now been thoroughly redrafted according to the reviewer’s comments

Round 2

Reviewer 3 Report

The authors answered correctly to all the points raised by by me. I am glad to accept the manuscript in its current form.